# Transcriptomics Using the Enriched Arabidopsis Shoot Apex Reveals Developmental Priming Genes Involved in Plastic Plant Growth under Salt Stress Conditions

**DOI:** 10.3390/plants11192546

**Published:** 2022-09-28

**Authors:** Ok-Kyoung Cha, Soeun Yang, Horim Lee

**Affiliations:** Department of Biotechnology, Duksung Women’s University, Seoul 01369, Korea

**Keywords:** developmental priming, RNA-seq, salt stress, shoot apical meristem, transcriptomics

## Abstract

In the shoot apical meristem (SAM), the homeostasis of the stem cell population supplying new cells for organ formation is likely a key mechanism of multicellular plant growth and development. As plants are sessile organisms and constantly encounter environmental abiotic stresses, postembryonic development from the shoot stem cell population must be considered with surrounding abiotic stresses for plant adaptation. However, the underlying molecular mechanisms for plant adaptation remain unclear. Previous studies found that the stem-cell-related mutant *clv3-2* has the property of salt tolerance without the differential response of typical stress-responsive genes compared to those in WT L*er*. Based on these facts, we hypothesized that shoot meristems contain developmental priming genes having comprehensively converged functions involved in abiotic stress response and development. To better understand the biological process of developmental priming genes in the SAM, we performed RNA sequencing (RNA-seq) and transcriptome analysis through comparing genome-wide gene expression profiles between enriched shoot apex and leaf tissues. As a result, 121 putative developmental priming genes differentially expressed in the shoot apex compared to the leaf were identified under normal and salt stress conditions. RNA-seq experiments also revealed the shoot apex-specific responsive genes for salt stress conditions. Based on combinatorial comparisons, 19 developmental priming genes were finally identified, including developmental genes related to cell division and abiotic/biotic-stress-responsive genes. Moreover, some priming genes showed CLV3-dependent responses under salt stress conditions in the *clv3-2*. These results presumably provide insight into how shoot meristem tissues have relatively high viability against stressful environmental conditions for the developmental plasticity of plants.

## 1. Introduction

In plants, the shoot apical meristem (SAM) activities, including cell division and differentiation, are regulated mainly by the CLAVATA3 peptide (CLV3p)-CLV receptor complex mediated signaling pathway, negatively inhibiting the WUSCHEL (WUS) transcription factor (TF) [1]. WUS proteins expressed in the organizing center cells move toward upper stem cells via plasmodesmata and activate *CLV3* expression directly in stem cells to promote the proliferation of undifferentiated stem cells [1]. The processed CLV3p are then secreted from stem cells to the neighboring cells and CLV receptor complexes, including CLV1 and BARELY ANY MERISTEM 1 (BAM1), recognize CLV3p ligands to induce downstream MAPK signaling to inhibit *WUS* expression [2]. These negative feedback loops between CLV3 and WUS eventually regulate the homeostasis of the stem cell population to control proper growth and postembryonic development of the aboveground lateral organs.

In nature, plants respond continuously to various environmental challenges during their lifespan because of a sessile property, and those abiotic/biotic stimuli usually affect plant growth and development. Therefore, plants must adapt to their growth, including endogenous developmental processes and morphological changes via their plastic properties in that even a single genotype can show a wide range of responses and phenotypes against unfavorable environmental stress conditions [3]. Developmental plasticity in plants is elaborately controlled by internal and external signals and one of the well-known examples is floral transition, which is determined by genetically integrated pathways such as developmental age, light length, vernalization, ambient temperature, and gibberellin (GA) [3]. In addition, flowering time is also plastically modulated by numerous abiotic/biotic stress signals, including salt, drought, heat, cold, nutrient, pathogens, and insects, for successful seed production [4]. Previously, it has been reported that salt stress delays flowering time with a dose-dependent manner through the repressive expression of *LEAFY* (*LFY*) in wild-type (WT) plants but not quadruple-DELLA mutants, suggesting that salt-stress-affected flowering time is mediated by the GA pathway [5]. Moreover, GA biosynthetic and signaling mutants showed enhanced plant growth via increased survival rate under toxic salt concentrations [5].

The aboveground lateral organs, including leaves, flowers, and stems, are supplied and developed from undifferentiated stem cells in the SAM. This indicates that the SAM is the core place of normal growth and development. Moreover, the SAM must have developmental plasticity for environmental stress stimuli. Indeed, several studies have reported plastic shoot growth-related stress responses. In *Rosa hybrida*, the drought stress stimulus reduced vegetative shoot growth and triggered the physiological defects of shoot length, weight, and floral organ formation in the reproductive phase [6]. The previous genetic approaches using natural accessions, such as Col-0 and Cvi-0, have shown that *ENHANCED SHOOT GROWTH UNDER MANNITOL STRESS 1/2* (*EGM1/2*) genes encoding putative receptor-like kinases are important for the shoot growth response to mannitol treatment [7]. The loss-of-function T-DNA mutants of *EGM1/2* genes in Col-0 and the Cvi-0 allele harboring natural mutations showed significantly better shoot growth under mannitol stress conditions. Another osmotic stress caused by a NaCl treatment has been reported to induce the abnormal proliferation of undifferentiated cells at the shoot apex in the loss-of-function double-mutant of *MSCL-LIKE 2* (*MSL2*) and *MSL3* genes encoding an MSL family of mechanosensitive ion channels [8]. In addition, a recent study reported that various abiotic stresses, including mannitol, sorbitol, NaCl, and hydrogen peroxide, affected physiological shoot growth in a highly dose-dependent manner [9].

Priming (also known as memory) is an adaptation strategy to gain tolerance against more severe stress conditions through the memorized effect induced by moderate stress for plastic growth and development of plants under unfavorable conditions [10,11]. In nature, mild stress can be manifested through various environmental fluctuations, such as dynamic and cyclic changes in temperature, water availability, and nutrient limitation [10]. The stress adaptation process by priming has been proposed to occur through the pre-accumulation of stress-responsive transcripts, the alteration in levels of key metabolites, and the accumulation and phosphorylation of stress mitogen-activated protein kinases (MPKs) and epigenetic regulation [10,11]. In other words, showing a priming effect can be speculated as a weak-stress state in plants. Interestingly, the expression of stress-inducible TF families, including abscisic-acid-responsive NAC (ANAC), WRKY, and basic leucine zipper (bZIP), was observed in undifferentiated stem cells under normal conditions [12]. Consistently, according to a recent report, the SAM intrinsically has hypoxic stress conditions along the apical-to-basal axis to regulate the production of lateral organs, indicating the stressful (also primed) state in the SAM [12,13]. Moreover, a recent study showed that the primary carbohydrate metabolism genes expressed under both mild and severe heat stress conditions are involved in the heat-stress memory at the SAM [14]. Although the various precedents strongly suggest that the potential role of the priming effect is essential for shoot growth against stress conditions, the molecular mechanisms involved in the stress priming in the SAM for developmental plasticity are poorly understood.

Here, RNA sequencing (RNA-seq) experiments were performed to identify higher or lower accumulated genes that are differentially expressed in the shoot apex compared to those in leaves under both normal and salt stress conditions. By analyzing RNA-seq results, we found 121 differentially expressed genes (DEGs) that displayed various criteria of gene ontology (GO) terms, such as development, abiotic/biotic stress response, phytohormone, and metabolism via GO enrichment analysis. RNA-seq experiments also revealed the shoot apex-specific responsive genes for salt stress conditions. Based on combinatorial comparisons, 19 developmental priming genes were finally identified, including the developmental genes related to cell division and abiotic/biotic-stress-responsive genes. Moreover, some priming genes showed CLV3-dependent responses under salt stress conditions in the *clv3-2*. These results show how shoot meristem tissues have a relatively high tolerance to stress conditions for the developmental plasticity of plants.

## 2. Results

### 2.1. Developmental Plasticity Involved in the Shoot Growth and Development against Salt Stress Conditions

Soil-grown mutant plants harboring deficiencies in shoot meristem development showed the salt-tolerant phenotype [15]. Consistent with this, seedling growth of shoot meristem-related mutants, such as *clv1 bam1* and *clv3-2* displayed enhanced survival rates under 100 and 150 mM NaCl conditions compared to the corresponding WT Columbia-0 (Col-0) and Landsberg *erecta* (L*er*), respectively (Figure 1). Because leucine-rich repeat-receptor-like kinases (LRR-RLKs) CLV1 and BAM1 act as receptors for CLV3p in the CLV–WUS negative feedback pathway regulating stem cell homeostasis in the SAM [1,2], these mutants usually display increased phenotypes of shoot meristem size and floral organ numbers, which are not related to stress tolerance. Therefore, this finding suggests that stem cell signaling consisting of peptide signals and receptors may function in abiotic resistance and maintenance in the SAM. In addition, WT L*er* showed more sensitive responses to shoot growth than another WT Col-0 under salt stress conditions (Figure 1). Although Col-0 seedlings showed reduced growth under 100 mM NaCl conditions compared to control conditions (Figure 1A), the survival rate was little affected (Figure 1B). In contrast, the survival rate, including defects of germination rate and shoot growth, was reduced significantly under 150 mM NaCl conditions (Figure 1B). Unlike Col-0, those defects of L*er* seedlings appeared under 100 mM NaCl conditions (Figure 1B). Consistently, it has been known that rosette size and ion leakage in L*er* were reduced and increased compared to those in Col-0, respectively, in the analysis using 160 Arabidopsis accessions under salt stress conditions [16]. These results suggest that the genetic information depending on Arabidopsis accessions is also essential for developmental plasticity on shoot growth against unfavorable salt stress conditions.

### 2.2. Identification of Developmental Priming Genes in the Shoot Apex

Because previous studies reported that canonical stress-responsive genes, including *KIN1*, *RD29A*, *RAB18*, and *DREB2A*, were not distinguishable between WT and *clv3-2* [15], we investigated the biological role of developmentally primed (intrinsically accumulated) genes acting as stress regulators in the shoot meristem compared to other leaf tissues by performing RNA sequencing (RNA-seq) to understand how shoot-stem-cell-related mutants show the salt-tolerant phenotype. For this, shoot apices and leaves of more salt-sensitive WT L*er* seedlings grown for nine days and treated without or with 200 mM NaCl for one day were harvested separately. A recent study showed that RNA-seq results using enriched shoot apices could represent SAM-specific tendency compared to those using functional zone/layer-specific cells [14,17].

Since heat-shock priming genes expressed in the SAM were previously found under moderate and severe stress conditions in RNA-seq analysis [14], we are also hypothesized that developmental priming genes are significantly expressed in the SAM than other tissues under both control (but primed, see the Introduction) and salt stress conditions. Therefore, we first isolated differentially expressed genes (DEGs) in the shoot apices compared to those in leaves under non-stress conditions in all three repetitive RNA-seq experiments (Figure 2A,B and Appendix A). Four hundred and seventy-three DEGs in the shoot apex were identified; 216 genes were upregulated, and 257 genes were downregulated (Figure 2A,B and Appendix A). Second, 484 DEGs in the shoot apex were isolated; 148 genes were upregulated, and 336 genes were downregulated under salt stress conditions (Figure 2C,D and Appendix A). Overall, 121 developmental priming genes were expressed significantly higher or lower in the shoot apex under both non-stress control and stress conditions (Figure 2E,F and Appendix A).

Gene Ontology (GO) enrichment analysis was performed using DAVID bioinformatics resources to determine the biological function of developmental priming genes in the SAM against salt stress conditions. Among the 121 DEGs, 70 and 51 genes were upregulated and downregulated, respectively, under control and stress conditions (Figure 2E). Based on the criteria of enrichment, 14 GO terms in biological process (BP) and 11 GO terms in the molecular function (MF) were identified in upregulated 70 DEGs (Figure 3A). An analysis of GO terms in biological processes included the following: auxin biosynthetic process (GO:0010601) with an enrichment value of 90.3-fold; nitrile biosynthetic process (GO:0080028) with 90.3-fold enrichment; cotyledon vascular tissue pattern formation (GO:0010588) with 35.1-fold enrichment; plasmodesmata-mediated intercellular transport (GO:0010497) with 31.6-fold enrichment; brassinosteroid homeostasis (GO:0010268) with 23.4-fold enrichment; regulation of cell proliferation (GO:0042127) with 22.6-fold enrichment; brassinosteroid biosynthetic process (GO:0016132) with 22.6-fold enrichment; glucosinolate catabolic process (GO:0019762) with 19.7-fold enrichment; mitotic cell cycle phase transition (GO:0044772) with 19.1-fold enrichment; microtubule-based movement (GO:0007018) with 18.9-fold enrichment; leaf morphogenesis (GO:0009965) with 14.6-fold enrichment; cell division (GO:0051301) with 7.8-fold enrichment; cell cycle (GO:0007049) with 5.7-fold enrichment; and multicellular organism development (GO:0007275) with 4.0-fold enrichment. These GO terms of upregulated priming genes in the shoot apex are mainly related to cell proliferation and hormone regulation (Appendix A). Consistent with the BP category, the highest value of GO enrichment in the MF category also revealed auxin-related IAA-amido synthetase activity (GO:0010279) with 106.7-fold enrichment, and transcription factor activity (GO:0003700) included various types of transcription factors involved in cell proliferation and auxin regulation (Table 1). Interestingly, several genes related to the regulation of phytohormones, such as auxin and brassinosteroid, were involved in plant growth and cell elongation/division by a KEGG enrichment analysis of upregulated DEGs in the shoot apex (Appendix A).

In the downregulated DEGs, 11 GO terms in BP and 9 GO terms in MF were identified (Figure 3B and Appendix A). In the BP category, enriched GO terms included the following: regulation of the defense response by callose (GO:2000071) with an enrichment value of 177.5-fold; response to oomycetes (GO:0002239) with 55.5-fold enrichment; SA mediated signaling pathway (GO:0009863) with 32.9-fold enrichment; cellular response to hypoxia (GO:0071456) with 7.3-fold enrichment; response to oxidative stress (GO:0006979) with 5.7-fold enrichment; defense response to fungus (GO:0050832) with 5.6-fold enrichment; defense response (GO:0006952) with 5.4-fold enrichment; signal transduction (GO:0007165) with 4.7-fold enrichment; defense response to bacteria (GO:0042742) with 4.4-fold enrichment; response to abscisic acid (GO:0009737) with 4.1-fold enrichment; and protein phosphorylation (GO:0006468) with 2.9-fold enrichment (Appendix A). The GO terms in BP were significantly related to biotic and abiotic stress responses. Analysis through the PANTHER classification system (http://www.pantherdb.org accessed on 22 September 2022) showed that even those genes included in the GO terms, such as signal transduction and protein phosphorylation, are primarily involved in the disease responses. Consistent with the function based on GO terms in the BP category, the GO terms of downregulated priming genes in the MF category mainly displayed the catalytic enzyme activities acting in the abiotic/biotic stress responses.

### 2.3. Clustering Analysis of Developmental Priming Genes in the Shoot Apex

K-Means/Medians Clustering analysis with average fold changes of 107 genes from 121 DEGs, expressed differentially in the shoot apex under both control and stress conditions, was performed to investigate the developmental priming gene more precisely. In this analysis, 14 genes showing no read (zero) value in the RNA-seq results were removed because of the unavailable fold ratio (Appendix A). Twelve of them were classified as an upregulated state in developmental priming genes. Seven encoded several types of transcription factors, including *KNOX ARABIDOPSIS THALIANA MEINOX* (*KNATM*), *SHOOT MERISTEMLESS* (*STM*), and *MYB117*, involved in regulating shoot and axillary meristem development [31,37,38].

As a result, two clustering groups showed increased or decreased patterns between control and NaCl conditions via the Pearson Correlation distance metric (Figure 4 and Appendix A). In the top 10 genes of increased DEGs in clustering analysis, most genes showed that increased patterns were changed in downregulated priming state. Those genes are involved in plant growth and biotic/abiotic stress responses (Table 2). In the top 10 genes of decreased DEGs in clustering analysis, the most dramatic change in the *GH3.3* and *ARABINOGALACTAN PROTEIN 1* (*AGP1*) genes acting in the regulation of salt tolerance and reproductive development occurred between the different primed states from upregulated to downregulated states (Table 2).

### 2.4. Identification of Salt-Inducible Genes Only in the Shoot Apex

The control and NaCl-treated RNA-seq results of shoot apex or leaf tissues were compared to find salt-inducible genes specifically enriched in the shoot apex. DEGs in the shoot apices treated with NaCl compared to those under control conditions in all three repetitive RNA-seq experiments were isolated (Figure 5A,B and Appendix A). Four hundred and ninety-nine differentially expressed genes were isolated in the shoot apex; 327 genes were upregulated by the NaCl treatment, and 172 genes were downregulated (Figure 5A,B and Appendix A). Four hundred and seventy-six DEGs were identified in the leaf tissue compared to those under control conditions; 377 genes were upregulated under salt stress conditions, and 99 genes were downregulated (Figure 5C,D and Appendix A). By comparing salt-inducible DEGs between the shoot apex and leaf, 379 preferentially enriched DEGs were found only in the shoot apex, 356 DEGs only in the leaf, and 120 DEGs in both the shoot apex and leaf tissues (Figure 5E). Interestingly, the GO terms related to development were revealed to the largest portion of preferentially enriched in the shoot apex (40%) but not those in the RNA-seq results of overlap (3%) and leaf (4%). In contrast, GO terms related to abiotic/biotic response showed the largest portion in overlap (58%) and leaf (44%) than the shoot apex (22%) (Figure 5F). The GO terms related to development in the shoot apex mainly included developmental processes involving cell division. Although all three categories contained GO terms related to leaf senescence (GO:0010150), those genes in the three categories were not overlapped, suggesting that they may function differently in developmental plasticity (Appendix A).

### 2.5. Identification of Developmental Priming Genes Related to Salt in the Shoot Apex

Based on the previous comparisons, Venn diagram analysis using developmental priming genes and salt-inducible DEGs revealed 117 developmental priming genes expressed more specifically in the shoot apex, of which 19 genes also showed salt-inducible changes only in the shoot apex (Figure 6A and Appendix A). This suggests that these 19 genes presumably play essential roles in developmental plasticity under salt stress conditions. According to GO enrichment analysis using DAVID, 98 genes displayed GO terms related to the biotic response, hormone regulation, organ development, and metabolic response, while 19 genes mainly showed GO terms related to the regulation of cell cycle/division (Figure 6B,C). Although 10 of the 19 genes were not included in GO enrichment analysis using DAVID, individual analysis of those genes revealed GO terms related to abiotic/biotic stress responses (Appendix A). Under control conditions, the genes associated with cell division and microtubule-based movement were included as upregulated priming genes in the shoot apex compared to other leaf tissue. By contrast, the salt treatment reduced the expression change of these genes. On the other hand, the expression of abiotic/biotic-stress-responsive genes associated with downregulated priming genes in the shoot apex under normal conditions was induced by salt stress conditions. These results revealed the functional behavior of developmental priming genes depending on salt stresses in the WT SAM.

Because previous studies showed that the *clv3-2* mutant exhibiting the increase in cell division or the decrease in cell differentiation has the property of salt resistance compared to WT L*er* via the non-canonical responses [15], qRT-PCR analysis was performed to verify the role of developmental priming genes in salt tolerance shown in *clv3-2*. As a result, some genes involved in cell division showed similar accumulation patterns in the shoot apex of L*er* and *clv3-2*, respectively (Figure 7A). In contrast, some genes classified as abiotic/biotic stress responses displayed abnormal responses only in *clv3-2* under salt stress conditions (Figure 7B). These results suggest that such genes involved in cell division may function in the recovery step of cell division after removing the stress stimuli instead of salt tolerance for plastic growth and development under stress conditions. In addition, they indicate that such genes involved in stress responses may play an important role in salt tolerance shown in *clv3-2*.

## 3. Discussion

In this study, we provided novel information on developmental priming genes associated with growth plasticity under unfavorable stress conditions using RNA-seq experiments and analysis. Developmental priming genes differentially expressed in the shoot apex under both control and salt stress conditions compared to leaf tissue contained approximately 10% TFs, including two bHLHs (*EGL3*, *At1g49830*), three homeodomain-containing proteins (*STM*, *ATHB13*, *ATHB33*), two C2H2 zinc finger proteins (*SGR5*, *ZFP2*), two MYB domain proteins (*AS1/ATPHAN*, *MYB117*), two WRKYs (*WRKY6*, *WRKY35*), an AP2 protein (*ANT*), and a basic leucine zipper (ATbZIP42) (Table 1). These types of TFs, including MYB, HD-ZIP, AP2, and WRKY, are known to mediate the transcriptional regulations in response to salt [53]. By clustering analysis of these TF genes, *ARABIDOPSIS THALIANA HOMEOBOX PROTEIN 33* (*ATHB33*)*/ZINC-FIGNER HOMEODOMAIN 5* (*ZHD5)*, *STM*, *ATHB13*, *ASYMMETRIC LEAVES 1* (*AS1*)*/PHANTASTICA* (*ATPHAN*)*,* and *AINTEGUMENTA* (*ANT*) genes were relatively related (Appendix A). The *ATHB33* gene has been reported to function in seed germination and root growth via negative regulation by ABA-induced *AUXIN RESPONSE FACTOR 2* (*ARF2*), and the overexpression of *ATHB33/ZHD*5 displayed accelerated plant growth, including floral architecture and leaf [25,26]. A recent study showed that *STM* expression is downregulated by the repressive class B ARFs, such as *ETTIN* (*ETT/ARF3*) and *ARF4,* to promote the initiation of flowers [21]. In addition to the developmental function of *STM*, recent reports showed that transgenic plants overexpressing *STM* adapt to drought stress via ABA-inducible *MYB96,* and the mechanical stress caused by curvature in the boundary domain of the shoot meristem induces *STM* expression correlated with auxin depletion [22,23]. *ATHB13* is also known to function in plant development of stem elongation, silique, seed production, and ovules via the regulation of cell division [18]. Because the activation of *ATHB13* showed disease resistance to powdery mildew species and tolerance to drought and salinity [19,20], these results may provide the role of *ATHB13* in the crosstalk among development and abiotic/biotic stress resistance. Interestingly, *SGR5* encoding zinc finger protein displayed the plastic adaptation role via the negative gravitropism of inflorescence stem under heat stress conditions [27]. In addition, *AS1/ATPHAN*, *ANT*, and *WRKY6* also showed multi-functional roles in development and abiotic/biotic stress responses (Table 1). Therefore, these various previous reports indicate the role of phytohormones, such as auxin and ABA for plant adaptation in the SAM, and suggest that the developmental priming genes, including TFs derived from RNA-seq experiments, may be reasonable candidates for the development and stress responses at the SAM.

After sensing salt stresses, plants can be placed in early and late responses. Immediate early responses are composed of calcium burst, ROS production, and activities of various pump/ion transporters, which play an essential role in ion homeostasis and sodium sequestering [19]. After early responses, the growth rate of plants changes toward the quiescent phase via the regulation of ABA levels and signaling, and the growth is then recovered through the raised levels of jasmonate (JA), brassinosteroid (BR), and gibberellic acid (GA) for homeostasis [54,55]. By salt application, microtubules are broken down in the early response phase, and the microtubule network is rebuilt dynamically during the quiescent phase [54,56]. Interestingly, in addition to TFs, other candidates as developmental priming genes were isolated in this study, and those putative functions based on GO enrichment analysis showed development-related biosynthesis and regulation of auxin, BR, and nitrile, as well as cell division and microtubule-based movement in the upregulated priming genes at the shoot apex (Figure 3A and Appendix A), whereas most downregulated priming genes contained ROS-related catalytic enzymes based on GO terms of molecular function (Figure 3B and Appendix A). Overall, these suggest that the putative developmental priming genes isolated from RNA-seq experiments probably represent early and late responsive genes after salt application.

A previous study of cell division in shoot meristems using EdU staining showed a decreased rate of cell division by a salt treatment and recovered cell division after removing salt stresses [15]. Because salt stress triggers defective cell division, we could also find various cell-division-related genes in the group of priming and salt-inducible DEGs in the shoot apex (Figure 5C and Appendix A). Interestingly, most genes classified as cell division in Appendix A function in the process between metaphase and anaphase via chromosome segregation during cell division. TPX2 (targeting protein for Xklp2) has been known to act as an activator of Aurora A kinase, which is one component of the spindle assembly checkpoint (SAC) complex delaying chromosome segregation in animals by sequestering CELL DIVISION CYCLE 20 (CDC20) [57,58]. In addition, TPX2 colocalized with AtAurora1 and γ-tubulin on spindle microtubules to control the meristematic cell proliferation in Arabidopsis [59]. CDC20.1, as a cofactor of the anaphase-promoting complex (APC), regulates Aurora localization and meiotic chromosome segregation, and the *cdc20.1* mutant showed defects of meiosis and male fertility [60]. In another salt priming candidate gene *CYC1BAT*, B-type cyclin and cdk1 form the mitosis promoting factor (MPF) complex phosphorylating TXP2 to activate Aurora kinase [61]. TOPII (topoisomerase II) is also important for chromosome stability, including chromosome segregation for DNA repair and mitosis/meiosis in eukaryotes [62,63]. However, since the expression patterns of these genes involved in the process of cell division under control and salt stress conditions in WT L*er* and *clv3-2* were almost similar (Figure 7A), the developmental priming genes related to cell division are probably not critical for the direct response against salt stimuli in *clv3-2* displaying salt tolerance compared to WT [15]. Instead, it seems to be important for resilience in plastic plant growth after salt application. Indeed, previous studies showed that the reduced rate of cell division under salt stress conditions in the SAM was almost recovered after removing the salt treatment [15]. Future studies will be needed to determine the biological function of these priming genes in regulating shoot development and stress responses in planta.

## 4. Materials and Methods

### 4.1. Plant Materials and Growth Conditions

The Arabidopsis thaliana accessions such as Columbia-0 (Col-0) and Landsberg *erecta* (L*er*) were used in this study as the wild-type. The genetic crossing for the *clv1 bam1* double mutant was reported previously [15]. For the analysis of survival rate under salt stresses, L*er*, *clv3-2*, Col-0, and *clv1 bam1* seeds were sowed on half MS plates containing 0.5 × MS, 0.5% Sucrose and 0.8% phytoagar without or with 100 or 150 mM NaCl, and seedlings in half MS plates were grown at 22–23 °C with 75 μmol m^−2^ s^−1^ light intensity under 16 h light/8 h dark photoperiod condition for 3 weeks. For RNA-seq and qRT-PCR experiments, L*er* and *clv3-2* seedlings were grown vertically on half MS plates for 9 days. Then, seedlings showing similar growth status were transferred to new half MS plates without or with 200 mM NaCl and grown for 1 day without any morphological change.

### 4.2. RNA Extraction and RNA Sequencing

Seedlings treated without or with NaCl treatment were dissected into shoot apex and leaves. Three biological replicates at each tissue with corresponding condition were used for RNA extraction, which was carried out with TRI Reagent (Molecular Research Center, Cincinnati, OH, USA) according to the manufacturer’s instructions. The libraries were prepared for 150 bp paired-end sequencing using TruSeq stranded mRNA Sample Preparation Kit (Illumina, CA, USA). mRNA molecules were purified and fragmented from 1 μg of total RNA using oligo (dT) magnetic beads. The fragmented mRNAs were synthesized as single-stranded cDNAs through random hexamer priming. By applying this as a template for second strand synthesis, double-stranded cDNA was prepared. After sequential process of end repair, A-tailing, and adapter ligation, cDNA libraries were amplified with PCR (Polymerase Chain Reaction). The quality of these cDNA libraries was evaluated with the Agilent 2100 BioAnalyzer (Agilent, CA, USA). They were quantified with the KAPA library quantification kit (Kapa Biosystems, MA, USA) according to the manufacturer’s library quantification protocol. Following cluster amplification of denatured templates, sequencing was progressed as paired-end (2 × 150 bp) using Illumina NovaSeq6000 (Illumina, San Diego, CA, USA). Raw data of RNA-seq reads obtained in this study were deposited with the accession nos. SRR20998325−SRR20998336 (BioProject: PRJNA868122) in the NCBI Short Read Archive (SRA) database.

### 4.3. Transcriptome Data Analysis

The adapter sequences and the ends of the reads less than Phred quality score 20 were trimmed, and simultaneously, reads shorter than 50 bp were removed by using cutadapt v.2.8 [64]. Filtered reads were mapped to the reference genome related to the species using the aligner STAR v.2.7.1a [65] following ENCODE standard options (refer to “Alignment” of “Help” section in the html report) with “-quantMode TranscriptomeSAM” option for estimation of transcriptome expression level. Gene expression estimation was performed by RSEM v.1.3.1 [66] considering the direction of the reads which are corresponding to the library protocol using option “-strandedness”. To improve the accuracy of the measurement, “-estimate-rspd” option was applied. All other options were set to default values. To normalize sequencing depth among samples, FPKM (fragments per kilobase of transcript per million) and TPM (transcripts per millions) values were calculated. Based on the estimated read counts in the previous step, DEGs were identified using the R package called TCC v.1.26.0 [67]. TCC package applies robust normalization strategies to compare tag count data. Normalization factors were calculated using the iterative DESeq2/edgeR method [68,69]. Q-value was calculated based on the *p*-value using the p.adjust function of R package with default parameter settings. The DEGs were identified based on the q value threshold less than 0.05 for correcting errors caused by multiple-testing [70]. DEGs were visualized by volcano, scatter, and MA plots. Heatmap and clustering analysis using the value of FPKM or the log2 ratio of fold change were visualized by the MeV 4.9.0 software (https://mev.tm4.org/#/about, accessed on 22 September 2022). The analysis of Gene Ontology (GO) enrichment and Kyoto Encyclopedia of Genes and Genomes (KEGG) for developmental priming genes or shoot apex-specific differentially expressed genes was performed using the DAVID software (https://david.ncifcrf.gov/, accessed on 22 September 2022), which is based on the database of Arabidopsis TAIR10 release.

### 4.4. cDNA Synthesis and qRT-PCR

First-stranded cDNAs from 1 μg of total RNA were synthesized using the ImProm-II^TM^ reverse transcriptase (Promega, Madison, WI, USA) according to the manufacturer’s instructions. Quantitative reverse transcription-polymerization chain reaction (qRT-qPCR) assays were performed with primers of developmental priming genes identified from RNA-seq experiments (Appendix A). Each reaction was carried out using MIC qPCR cycler (Bio Molecular Systems, Upper Coomera, QLD, Australia) with SYBR Green Realtime PCR Master Mix (TOYOBO, Osaka, Japan).

### 4.5. Statistical Analyses

All quantitative results were analyzed by one-way analysis of variation (ANOVA) using SPSS (ver. 27) statistic software. Significant differences and different letters were determined by one-way ANOVA with post-hoc Turkey’s HSD multiple comparison tests.

## 5. Conclusions

This study provides a fundamental resource for understanding the developmental priming effect in shoot meristems orchestrating the aboveground growth. Under normal (non-stress) conditions, the putative priming genes regulating phytohormones, development, and abiotic/biotic stress responses are expressed differently in the shoot apex than in leaf tissues. Because many stress-related genes are highly or lowly pre-accumulated in the shoot apex, the transcriptome data also support the idea of a primed state in the SAM to respond efficiently to incoming severe stress conditions [12,13]. In addition, some cell division and stress-responsive genes are probably associated with a wide range of salt stress responses for plastic growth and development under salt stress conditions. Because the defective cell division within the shoot meristems affects overall growth and development, these developmental priming candidate genes isolated from RNA-seq experiments using enriched shoot apex may provide new insight into understanding plastic growth adaptation under salt stress conditions.

## Figures and Tables

**Figure 1 plants-11-02546-f001:**
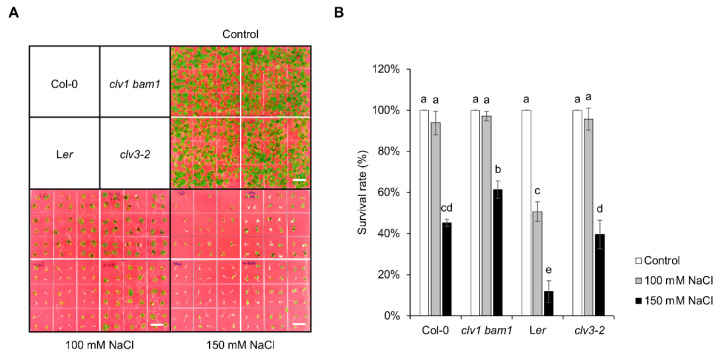
Stem-cell-related mutants, such as *clv1 bam1* and *clv3-2*, show enhanced shoot growth phenotypes under salt stress conditions. (**A**) Salt-tolerant phenotypes of WT Col-0, *clv1 bam1*, WT L*er*, and *clv3-2*. Seedlings were grown on half MS plates without (Control) or with 100 or 150 mM NaCl for 3 weeks; Scale bars, 1 cm. (**B**) Survival rate of Col-0, *clv1 bam1*, L*er*, and *clv3-2* seedlings. The data are presented as the mean ± SD (*n* = 4). The different letters indicate a significant difference (*p* < 0.05) according to the one-way ANOVA with post-hoc Turkey’s HSD test.

**Figure 2 plants-11-02546-f002:**
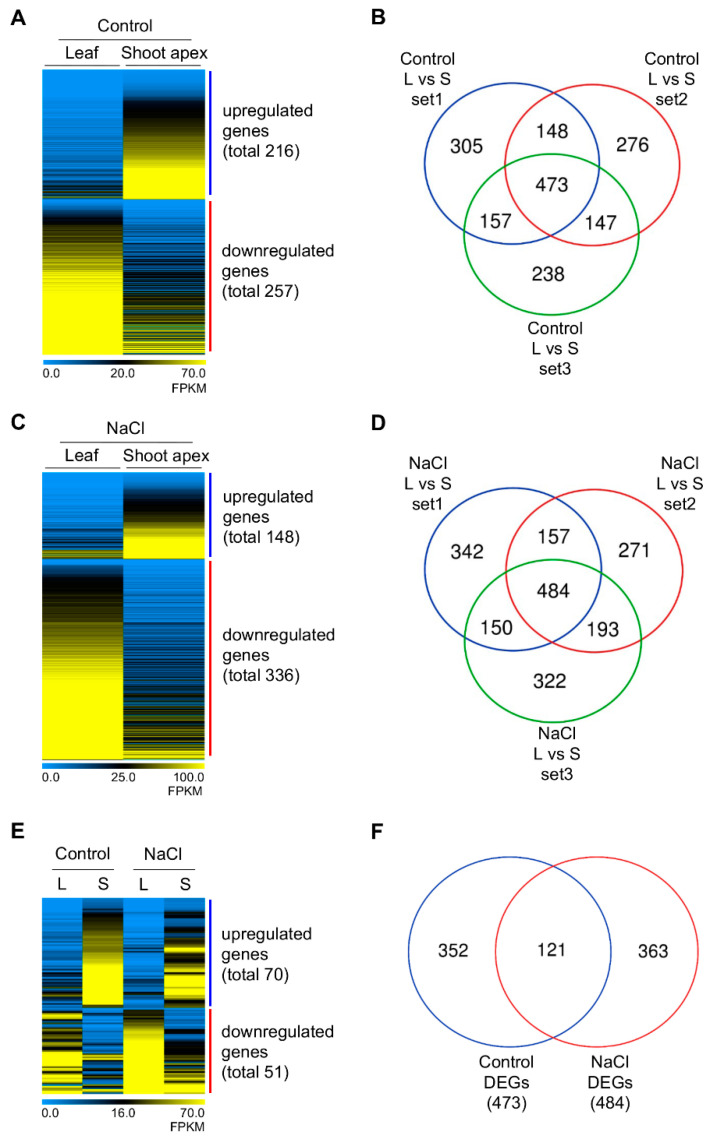
Expression profiles of developmental priming genes in the shoot apex. Differentially expressed genes (DEGs) of three replicates (set 1, 2, 3) were visualized by Heatmap using average FPKM values (**A**,**C**,**E**) and Venn diagram (**B**,**D**,**F**); Developmental DEGs in the shoot apex compared to those in the leaf tissue under control (**A**,**B**) or NaCl conditions (**C**,**D**); Overlapped DEGs in the shoot apex under both conditions (**E**,**F**); Heatmap is the result of hierarchical clustering via MeV 4.9.0 with the current metric Covariance; L, leaf; S, shoot apex.

**Figure 3 plants-11-02546-f003:**
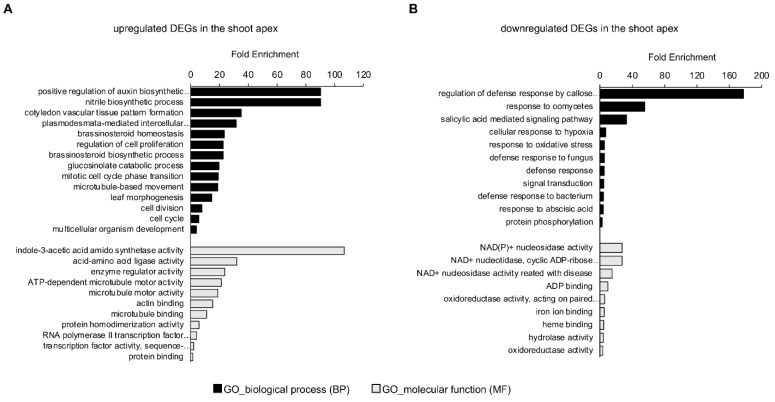
Gene Ontology (GO) enrichment analysis of developmental priming genes in the shoot apex. GO enrichment analysis of upregulated DEGs (**A**) and downregulated DEGs (**B**) in the shoot apex; Biological process (BP) and molecular function (MF) of developmental priming genes were determined via the DAVID program; Selected genes of *p*-value < 0.05 following the test were regarded it as statistically significant.

**Figure 4 plants-11-02546-f004:**
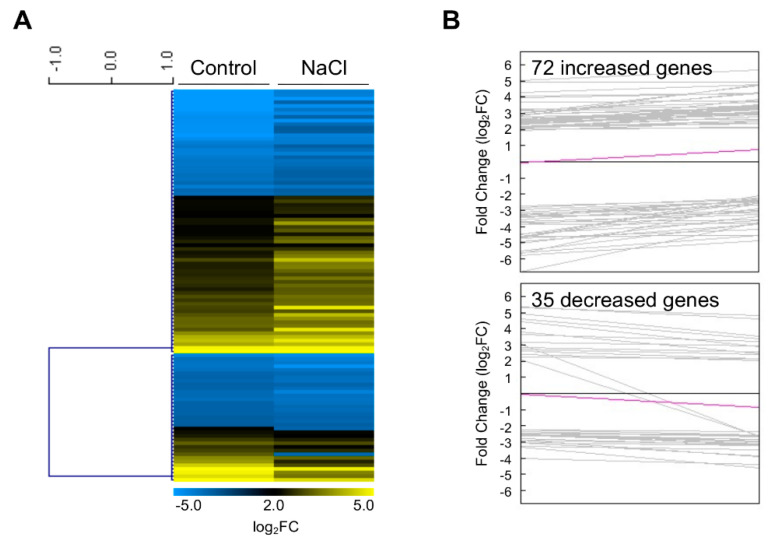
Clustering analysis of developmental priming genes in the shoot apex. (**A**) Fold changes of 107 genes from 121 DEGs were visualized by Heatmap after analyzing K−Means/K−Medians clustering with the current metric Pearson Correlation. Fold change was the average value of three replicates. (**B**) Profiles of increased (upper) and decreased (lower) genes. The bold pink line represents the average profile.

**Figure 5 plants-11-02546-f005:**
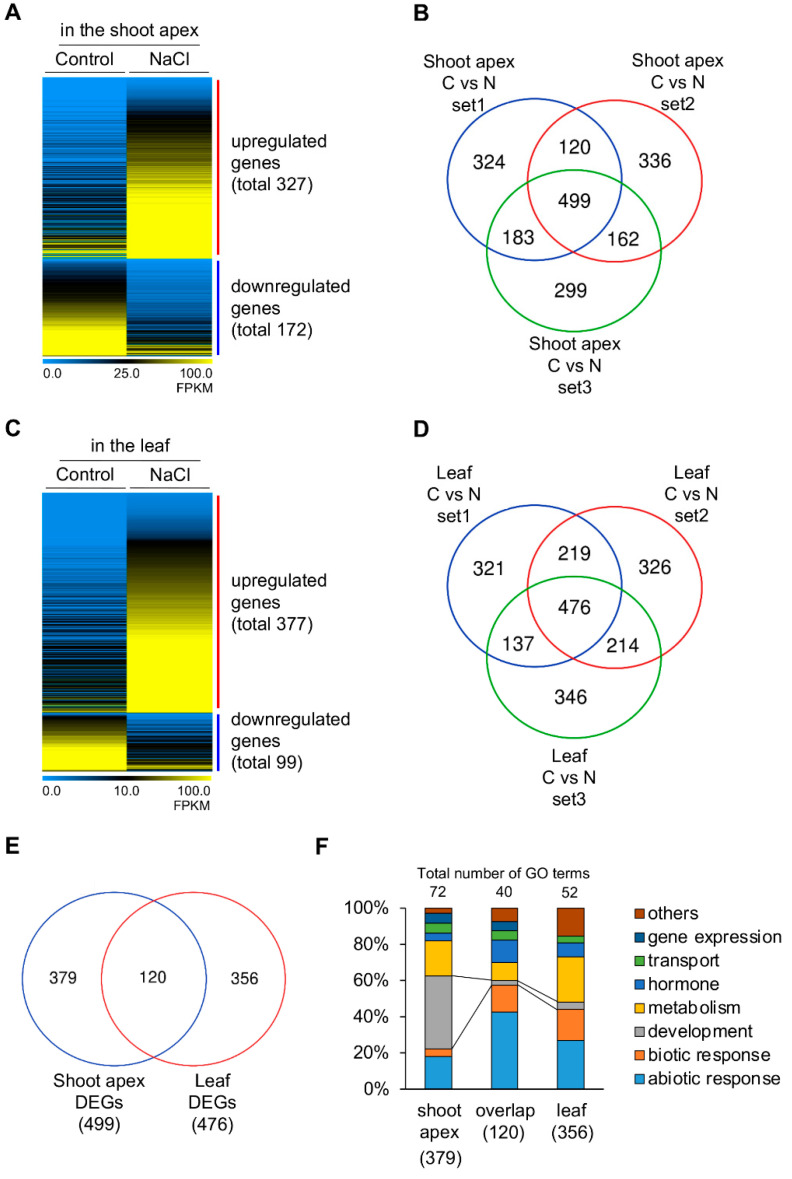
Expression profiles of shoot apex-specific DEGs. Specifically expressed genes in the shoot apex or the leaf tissue in response to salt stress conditions were visualized by Heatmap using average FPKM values (**A**,**C**) and Venn diagram (**B**,**D**,**E**); (**A**,**B**) Specifically expressed genes of three replicates (set 1, 2, 3) in the shoot apex by NaCl treatment (N) compared to control (C); (**C**,**D**) Specifically expressed genes of three replicates in the leaf tissue by NaCl treatment compared to control; (**E**) Overlapped Venn diagram of salt inducible DEGs between shoot apex and leaf; (**F**) Distributions of GO terms of salt inducible DEGs in the shoot apex, overlap, and leaf; Heatmap is the result of hierarchical clustering via MeV 4.9.0 with the current metric Covariance; C, Control; N, NaCl.

**Figure 6 plants-11-02546-f006:**
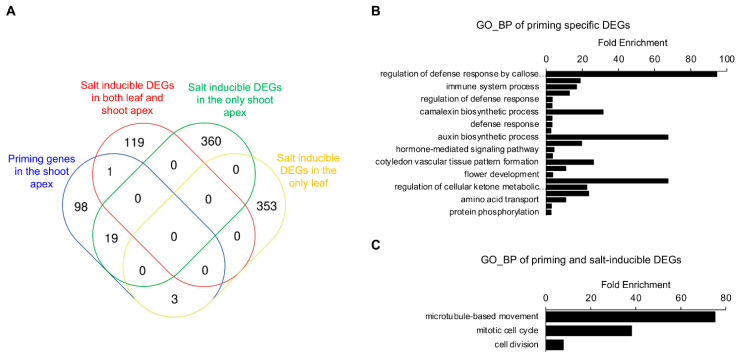
Shoot apex-specific developmental priming genes in response to salt stress. (**A**) Combinatorial comparisons of developmental priming (shoot apex) and three kinds of salt inducible genes (shoot apex, leaf and both) via Venn diagram; (**B**) GO enrichment analysis of developmental priming genes regardless of salt stress; (**C**) GO enrichment analysis of developmental priming genes coincident with salt stress; (**B**,**C**) GO enrichment analysis was based on the biological process (BP) and selected genes of *p*-value < 0.05 following the test were regarded it as statistically significant.

**Figure 7 plants-11-02546-f007:**
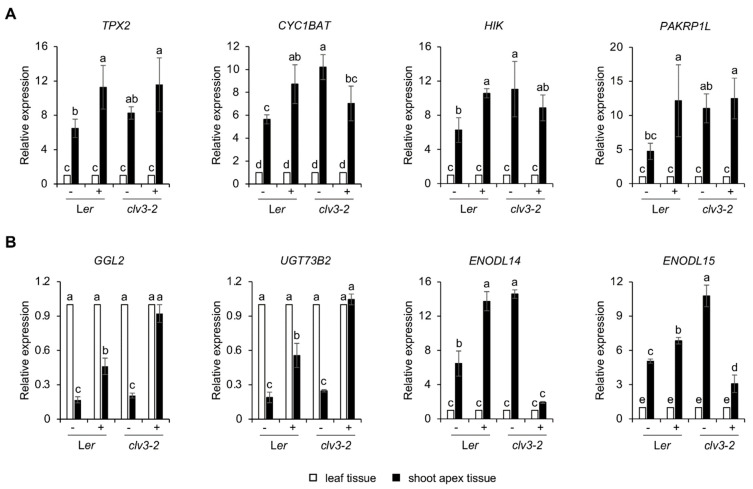
Expression patterns of developmental priming genes in WT L*er* and *clv3-2* shoot apices without (−) or with (+) salt stress conditions. Developmental priming genes related to cell division (**A**) and abiotic/biotic stress responses (**B**) were examined by qRT-PCR and *ACT2* genes were used to normalize each gene expression. The data are presented as the mean ± SD (*n* = 3~5). The different letters indicate a significant difference (*p* < 0.05) according to the one-way ANOVA with post-hoc Turkey’s HSD test.

**Table 1 plants-11-02546-t001:** Developmental priming TF genes preferentially expressed in the shoot apex.

AGI Number	Synonym	Type of TF	Developmental Function	Stress Function	References
*AT3G30530*	*ATbZIP42*	basic leucine zipper	-	-	
*AT1G69780*	*ATHB13*	HD-ZIP family	reproductive organ development	low temperature/drought/salt/disease	[18,19,20]
*AT1G62360*	*STM*	KNOX/ELK	SAM initiation/maintenance via auxin	mechanical/drought stress response	[21,22,23,24]
*AT1G75240*	*ATHB33*	ZF-HD family	root and whole plant growth	ABA response	[25,26]
*AT2G01940*	*SGR5*	zinc finger protein	shoot gravitropism	high temperature response	[27]
*AT5G57520*	*ZFP2*	zinc finger protein	abscission of floral organ	-	[28]
*AT2G37630*	*AS1*	MYB protein	abscission of floral organ	defense response	[29,30]
*AT1G26780*	*MYB117*	MYB protein	axillary meristem formation	-	[31]
*AT4G37750*	*ANT*	AP2 family protein	cell proliferation in root growth	defense response	[32,33]
*AT2G34830*	*WRKY35*	WRKY protein	-	-	
*AT1G62300*	*WRKY6*	WRKY protein	ABA signaling in seedling development	low Pi stress response	[34,35]
*AT1G63650*	*EGL3*	bHLH family	cell patterning in root epidermis	-	[36]
*AT1G49830*	-	bHLH family	-	-	

**Table 2 plants-11-02546-t002:** List of clustered genes.

AGI Number	Synonym	Control			NaCl			Function	References
		Avg. log_2_FC	SD	Primed State	Avg. log_2_FC	SD	Primed State		
**increased DEGs**									
*AT4G10120*	*ATSPS4F*	−5.67	0.36	DOWN	−2.50	0.33	DOWN	osmotic stress	[39]
*AT5G64120*	*PER71*	−6.79	0.79	DOWN	−3.76	0.78	DOWN	negative effects of growth via ROS	[40]
*AT3G48310*	*CYP71A22*	−4.64	0.22	DOWN	−2.10	0.36	DOWN	-	
*AT3G22060*	*CRRSP38*	−4.71	0.12	DOWN	−2.26	0.47	DOWN	regulation of leaf growth	[41]
*AT5G20230*	*ATBCB*	−4.47	0.56	DOWN	−2.48	0.54	DOWN	cold acclimation in lignin biosynthesis	[42]
*AT3G16400*	*NSP1*	2.84	0.12	UP	4.76	0.79	UP	defense metabolite formation	[43]
*AT3G16410*	*NSP4*	2.99	0.40	UP	4.73	0.41	UP	defense metabolite formation	[43]
*AT2G26560*	*PLP2*	−5.49	1.15	DOWN	−3.77	1.41	DOWN	cell death and biotic stress response	[44]
*AT2G39200*	*MLO12*	−5.05	0.54	DOWN	−3.35	0.72	DOWN	defense response	[45]
*AT4G08870*	*ARGAH2*	−4.60	0.26	DOWN	−2.90	1.14	DOWN	abiotic stress tolerance	[46]
**decreased DEGs**									
*AT2G23170*	*GH3.3*	3.06	0.56	UP	−2.67	0.96	DOWN	IAA-dependent salt tolerance	[47]
*AT5G64310*	*AGP1*	2.08	0.06	UP	−2.64	0.37	DOWN	reproductive process	[48]
*AT3G23430*	*PHO1*	4.44	0.38	UP	2.89	0.12	UP	shoot phosphate level and growth	[49]
*AT5G23940*	*EMB3009*	4.90	0.65	UP	3.55	0.53	UP	cutin Biosynthesis	[50]
*AT4G27260*	*GH3.5*	3.81	0.21	UP	2.47	0.29	UP	IAA-dependent salt tolerance	[47]
*AT5G40780*	*LHT1*	−3.33	0.32	DOWN	−4.62	0.41	DOWN	uptake of amino acid	[51]
*AT1G75780*	*TUB1*	4.61	0.48	UP	3.36	0.44	UP	-	
*AT1G62300*	*WRKY6*	−2.89	0.16	DOWN	−3.91	0.49	DOWN	low-Pi stress response	[35]
*AT3G01290*	*HIR2*	−3.07	0.48	DOWN	−3.83	0.55	DOWN	hypersensitive response with immunity	[52]
*AT2G28790*	-	5.33	0.33	UP	4.61	0.27	UP	-	-

## Data Availability

RNA-seq data analyzed in this study are available in the NCBI SRA database (SRR20998325−SRR20998336).

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
