# Peer review of "Transcriptomics Using the Enriched Arabidopsis Shoot Apex Reveals Developmental Priming Genes Involved in Plastic Plant Growth under Salt Stress Conditions"

_plants, 2022, doi:10.3390/plants11192546_

Round 1
Reviewer 2 Report
Salinity stress is becoming a big threat to plants. Recent studies focused on the improvement of plant tolerance to abiotic stresses. In the manuscript, the authors performed transcriptomics to analyze differentially expressed genes in shoot apical meristem, as compared to those in leaf tissue, under normal growth conditions and salt stress conditions. The data showed that several developmental priming candidate genes were identified in shoot apical meristem, suggesting a novel role of shoot apical meristem in regulating plant response to salt stresses.
In sum, the manuscript was generally well organized and well written. Moreover, the authors present a very interesting topic. There are currently many interests in the function of shoot apical meristem on plant tolerance against saline stress in plants. It would be of broad interest to the plant community, and the “Plants” readers. However, I have some concerns about the manuscript, before publication:
1. In Figure 2, the authors present all three biological replicates from different tissue samples (leaf or shoot apical) in heat map. Usually, the authors should present the average date from the several biological replicates, but not the single replicates. In my opinion, the authors need to present the average data in the different tissue samples, which is easy for readers to understand the results. There is a similar problem in Figure 4.
2. The resolution of Figures 2, 3, 4, and 5 are not good. I print the manuscript and the figure is very difficult to read. Please increase the resolution of figure 2, 3, 4 and 5.
3. As the abbreviation was used for the first time, the authors should explain it. Such as on page 2 line 56 “EGM1/2”. Please check all of the abbreviations in the manuscript.
4. In the “result” section, in the first part, the authors used the two shoot meristem-related mutants, such as clv1 bam1, and clv3-2. Some necessary background information needs to be provided, and the authors need to explain the reasons why these two mutants were selected in this study.
Round 2
Reviewer 2 Report
There are some outstanding improvements in the revised manuscript. In addition, I appreciated that the authors answered my concerns one by one. Now, the figures look much better and more professional. Also, the authors already submit the raw data of RNA-seq to an open access database that the readers can find. To sum up, the manuscript was organized and well-written. Therefore, I suggested the manuscript is ready to be published.